# Follow-Up of Eight-Weeks Detraining Period after Exercise Program on Health Profiles of Older Women

**DOI:** 10.3390/healthcare11233021

**Published:** 2023-11-22

**Authors:** Luís Leitão, Yuri Campos, Ana Cristina Corrêa Figueira, Teresa Figueiredo, Ana Pereira

**Affiliations:** 1Sciences and Technology Department, Superior School of Education of Polytechnic Institute of Setubal, 2910-761 Setúbal, Portugal; ana.figueira@ese.ips.pt (A.C.C.F.); teresa.figueiredo@ese.ips.pt (T.F.);; 2Life Quality Research Centre, 2400-901 Leiria, Portugal; 3Post Graduate Program in Physical Education, Federal University of Juiz de Fora, Juiz de Fora 36036-900, Brazil; 4Study Group and Research in Neuromuscular Responses, Federal University of Lavras, Lavras 37200-900, Brazil

**Keywords:** aging, functional capacity, lipid profile, multicomponent training

## Abstract

Background: The multicomponent training program (MTP) is a physical exercise strategy used to combat the sedentary lifestyle in older women (OW). However, periods of interruption in training are common in this population. The aim of our study was to analyze the 8-week MTP effects followed by two, four, and eight weeks of interruption on the lipid profile (LP) and functional capacity (FC) of OW. Methods: Twenty-one OW (experimental group [EG], 67.6 ± 3.1 years; 1.55 ± 0.35 m) were subjected to an 8-week MTP followed by a detraining period, and 14 OW (control group [CG], 69.4 ± 4.7 years; 1.61 ± 0.26 m) maintained their daily routine. FC (i.e., 30-s chair stand [30-CS], 8-foot up and go [8-FUG], 6-min walk [6-MWT], handgrip strength [HGS], and heart-rate peak during 6-WMT [HRPeak]), total cholesterol (TC) and triglycerides (TG) were assessed before and after MTP and two, four, and eight weeks after MTP. Results: 8-week MTP resulted in higher FC and decreased LP values in EG (*p <* 0.05); two and four weeks of detraining did not promote changes. After eight weeks of detraining TC (ES: 2.74; *p* = 0.00), TG (ES: 1.93; *p* = 0.00), HGS (ES: 0.49, *p* = 0.00), HRPeak (ES: 1.01, *p* = 0.00), 6-MWT (ES: 0.54, *p* = 0.04), and 8-FUG (ES: 1.20, *p* = 0.01) declined significantly. Conclusions: Periods of more than four weeks of detraining should be avoided to promote a good quality of life and health in OW. If older people interrupt training for a period longer than four weeks, physical-education professionals must outline specific training strategies to maintain the adaptations acquired with MTP. Future studies should establish these criteria based on ideal training volume, intensity, and frequency.

## 1. Introduction

Cardiovascular diseases are considered by the World Health Organization as the most common cause of death in the world’s population [1], and one of its main risk factors is the so-called dyslipidemia [2]. Dyslipidemia is characterized by an abnormal lipid profile (LP) in the bloodstream [3], and is usually associated with elevated levels of total cholesterol (TC), low-density lipoprotein cholesterol, and triglycerides (TG), as well as reduced levels of high-density lipoprotein cholesterol (HDL-C) [4], which plays an important role as the “good” cholesterol in protecting against death from cardiovascular disease [5]. Within these characteristics, hyperlipidemia is the most common form of dyslipidemia in the older population, and considering that the world population is moving toward an “aged society” [6], with estimated increases in the number of individuals over 65 years of age rising from 7.9% in 2010 to 17% in 2050 [7,8], specific strategies are needed to manage modifiable cardiovascular risk factors [9].

There is some controversy regarding high serum lipids as a risk factor for mortality among older people, and some studies have shown that increased serum lipids and lipoproteins have been associated with increased risks of all-cause or cardiovascular mortality in older people [10,11,12]; however, other studies have not shown this association [13,14]. Nonetheless, there is no doubt that physical exercise may improve the quality of life [15], cardiovascular [16], musculoskeletal [17], and cognitive health [18], minimizing cardiovascular risk and mortality in older people [19].

One of the most used training strategies to improve cardiovascular health [16] and physical functioning [20] in older adults is the multicomponent training program (MTP) [21]. Bouaziz et al. [22] observed improvements in cardiovascular health after MTP in older people. Improvements of around 10% to 20.0% were observed in the maximal oxygen consumption peak, in addition to significant reductions of 3.0% and 4.0% in fat mass, as well as a significant reduction of 6.0% to 8.2% in TG and HDL-C serum levels. With regard to functional capacity (FC), a recent literature review with meta-analysis demonstrated the effectiveness of MTP in improving parameters related to walking speed, short physical performance battery, and upper- and lower-limb strength, as well as the timed up and go test [23]. MTP consists of a combination of aerobic, strength, and flexibility exercises, and is indicated in the long term for older adults [24]. To maintain the positive effects of MTP, older adults should continue with a workout routine for as long as possible [25]. However, older adults are more vulnerable to interrupting a workout routine because of physical and behavioral factors [26], as well as seasonality [27]. In this context, it is important to better understand the short- and long-term effects and residual effect after cessation of a workout routine [28].

Detraining can be considered the partial or total interruption of a workout routine, or the partial or total loss of benefits acquired in response to an insufficient training stimulus [29]. Some studies have found a decline in the FC of older people over a period of three to five months [30,31,32,33]. However, as far as we know, only one study has verified the loss of FC in short periods (i.e., 2-, 4-, and 6-week) [34]. Older people have their workout routines interrupted due to falls, illness, and hospitalization [35]. Another preponderant factor for the physical-training interruption in older people is the vacation period [31]; it is important, therefore, to know these short-term losses so that physical-education professionals can devise appropriate strategies to minimize them. In addition, although studies have been dedicated to verifying the loss of FC during detraining, to our knowledge only a few studies considered the LP assessment [21,29,36], an important indicator of cardiovascular risk. Therefore, the aim of our study was to analyze the MTP effects followed by two, four, and eight weeks of detraining on the LP and FC of older women (OW).

## 2. Materials and Methods

### 2.1. Sample

Forty-one community-dwelling Caucasian OW with high levels of TC and TG, and functionally independent, volunteered to participate in this study. Before the beginning, all the participants underwent a medical assessment to participate in the training program. The exclusion criteria were: (a) to have already participated in any regular physical-activity program training in the last 12 months; (b) to present osteoarticular and/or muscular dysfunction that could interfere with the execution of the proposed motion; (c) to have heart problems where the exercise prescription would injure the health of an OW, and (d) a medical/mental contraindication. Before the data collection, participants were informed about the study procedures and signed an informed consent that was written by the researcher who conducted and designed the study. The study was approved by the local institutional ethics committee (protocol number 2,887,652). All participants signed an informed consent form, and the study was conducted in compliance with the Declaration of Helsinki. They were advised to maintain their previous lifestyle throughout the study, including dietary patterns and daily routine. Coffee, tea, alcohol, tobacco consumption, and strenuous exercise were prohibited 24 h before the experimental procedures.

### 2.2. Procedures

Using simple random sampling, the participants were separated into two groups: the experimental (EG, *n* = 21; 67.6 ± 3.1 years; 1.55 ± 0.35 m) group, who performed an 8-week MTP followed by two, four, and eight weeks of detraining; and the control group (CG: *n* = 14; 69.4 ± 4.7 years; 1.61 ± 0.66 m) group, who maintained their daily routine. In the CG, six older women were excluded because they did not attend all assessments.

Data were collected by the same team of researchers under the same environmental conditions (morning between 10:00 and 12:00 hours at 22 and 24 °C and 55–65% humidity). The same instruments and materials were used at all times to measure anthropometric parameters (weight; eight; BMI), heart-rate peak during a 6-min walk test, TC and TG, and functional performance (agility, cardiorespiratory fitness, lower body strength, and handgrip). These variables were measured before and after the 8-week MTP and after two, four, and eight weeks of detraining.

#### 2.2.1. Multicomponent Training Program (MTP)

MTP consisted of 45-min, three times per week for eight consecutive weeks, totaling 24 sessions. The design and prescription of the exercise program were conducted by a physical-education and exercise specialist in training for older adults according to the ACSM guidelines for exercise prescription [21,37]. Each training session consisted of aerobic and muscle-resistance training where, before the start of the session, a warm-up activity was performed (i.e., slow walking, calisthenics, and stretching exercises), and then after the end of the session, a cool-down was performed (i.e., static and dynamic stretching techniques). In all the sessions, music was used to motivate and promote social well-being among the participants.

For the cardiorespiratory workout routine, we used exercises with aerobic choreography, the intensity was maintained between 2 and 3 in accordance with Borg’s perceived exertion scale (RPE) adapted in the first four weeks and gradually increased to 4–5 every two weeks. Therefore, in the muscular phase, exercises for the upper and lower limbs were performed in a circuit with one’s own body weight, adjacent to agility, mobility, and coordination exercises with 20 to 30 s rest between sets (e.g.,: air squat; arm raise; heel to toe walking; obstacle overpass; back leg raises). The intensity was progressive, especially in the first four weeks of training, to allow adequate familiarization with the exercises and the correct and safe execution and breathing technique, and after this period we increased the series and repetitions from two to four and from 16 to 30, respectively [21].

#### 2.2.2. Detraining Period (DT)

DT follow-up occurred at two, four, and eight weeks after MTP. All participants were instructed to continue their daily activities and maintain their eating patterns, but to avoid any type of regular physical exercise. To control these issues, all the participants were systematically contacted by the researchers to ensure that they were complying with the DT requirements.

#### 2.2.3. Anthropometric Variables

A scale (OMRON BF 303, Matsusaka, Japan), a stadiometer (Seca, Hamburg, Germany), and a bioelectrical impedance analysis were used for body mass (kg), height (cm), and body fat percentage (%BF), respectively.

#### 2.2.4. Lipidic Profile

Dyslipidemia was evaluated by the TC (mg/dL) and TG (mg/dL) using a Cobas Accutrend Plus (Roche Diagnostics GmbH, Mannheim, Germany). The blood collection was used with the pen puncture Accu-Chek Softclix^®^ Pro, on the distal phalanx palmar of the third finger of the right hand.

#### 2.2.5. Functional Capacity Battery Test

After a 10 min of warm-up of physical exercises, the functional capacity battery test (FC) was performed according to Rikli & Jones [38]. Before and after the eight-week MTP and also after two, four, and eight weeks of DT. All the participants performed three motor tests:(1)lower-limb strength (30-s chair stand, 30-CS): number of full stands a subject can complete within 30 s starting from the chair.(2)agility/dynamic balance (eight foot up and go, eight FUGs): Time clocked from the moment that the subject gets up from the chair and walks eight feet to and around a cone, and returns to the chair.(3)aerobic capacity (6-min walk, 6-MWT): number of meters performed walking as fast as possible during six minutes around a 50-m stretch of unimpeded walkway.

Furthermore, an additional test was used, the handgrip strength test (HGS). The dominant hand was assessed with a manual hydraulic dynamometer (Jamar 5030J1, Jamar Technologies, Horsham, PA, USA). The subject performed the test standing, with the elbow in full extension, and the upper extremity at a 90-degree internal rotation parallel to the largest axis of the body, with the grip width in the second position. For this test, the highest value of three records was employed [39]. During the 6-MWT we also registered the heart-rate peak (HRPeak) with the Garmin HRM-RUN strap.

### 2.3. Statistical Analysis

The data were analyzed with SPSS 19.0 for Windows (SPSS Inc., Chicago, IL, USA). All data were expressed as mean ± standard deviation with a 2-tailed *p* value of ≤0.05 required for significance. Descriptive procedures of central tendency and dispersion were used to characterize the variable’s values, and the normality of our sample was verified by the Shapiro–Wilk test. For inferential data analysis, a repeated measures ANOVA was used to compare the mean values of each variable at each time of the study, followed by Bonferroni’s post-hoc test. The sphericity assumption was verified through Mauchly’s test. The meaningfulness of the outcomes was estimated through the effect size (ES, Cohen’s d, means divided by the standard deviation): 0.2 or less is a small ES, about 0.5 is a moderate ES, and 0.8 or more is a large ES. The delta percentage (∆%) was calculated via the standard formula: ∆% = [(posttest score − pretest score)/pretest score] × 100. A sample size calculation was made with the G*Power program considering a 0.3 effect size, 83% of power and *p <* 0.05 of significance, a sample of 21 older women per group would be necessary.

## 3. Results

Table 1 and Table 2 show the response of OW to an 8-week MTP followed by an 8-week DT. After the exercise program, TC (ES: 5.07; *p* = 0.01), TG (ES: 4.02; *p* = 0.01), HGS (ES: 0.78, *p* = 0.01), HRPeak (ES: 1.43, *p* = 0.01), 6-MWT (ES: 0.48, *p* = 0.03), 8-FUG (ES: 0.07, *p* = 0.04), and 30-CS (ES: 0.63, *p* = 0.03) improved significantly. The CG did not change. The EG adherence to MTP was 91%.

The follow-up DT showed that the first two and four weeks did not lead to any changes in the LP or FC, and only BF increased after four weeks (*p <* 0.05). At the end of the eighth week of DT, there were negative changes in TC (ES: 2.74; *p* = 0.00), TG (ES: 1.93; *p* = 0.00), HGS (ES: 0.49, *p* = 0.00), HRPeak (ES: 1.01, *p* = 0.00), 6-MWT (ES: 0.54, *p* = 0.04), and 8-FUG (ES: 1.2, *p* = 0.01), except in 30-CS (ES: 0.46, *p* = 0.10). Comparing the end of DT with the baseline values, we found that TC (ES: 2.73, *p* = 0.01), TG (ES: 1.56, *p* = 0.01), and HGS (ES: 0.28, *p* = 0.01) had improved.

## 4. Discussion

The aim of the present study was to analyze the MTP effects followed by two, four, and eight weeks of detraining on the LP and FC of OW. The main results of the study showed that 8-week MTP significantly improved the parameters related to LP (i.e., TC and TG), as well as the parameters associated with FC (i.e., HGS, HRPeak, 6-MWT, 8-FUG, and 30-CS) for EG regarding baseline, except for the %BF variable. In contrast, the CG did not change their parameters after the 8-week. Furthermore, the LP showed a significant decline in serum TC and TG levels with 8-week DT when compared to 8-week MTP. %BF also demonstrated a significant worsening between 8-week MTP and 4-week DT. Likewise, the parameters attributed to FC (i.e., HGS, HRPeak, 6-MWT, and 8-FUG) in EG also showed a significant decline in the 8-week DT compared to the 8-week MTP, except for 30-CS.

It is well documented that the aging process is accompanied by multiple systemic dysfunctions in the body, as well as disturbances in the lipid metabolism and a chronic inflammatory state that contributes to the onset of cardiovascular diseases [40,41]. In this sense, scoping reviews [16], including meta-analyses [3], have shown the role of physical exercise in controlling the LP through the decrease in TC, TG, and low-density lipoprotein and an increase in high-density lipoprotein. Reviews involving strength training [42] and aerobic training [40] have suggested a positive effect of both types of exercise on improving LP, including the parameters assessed in the present study (i.e., TC and TG). Thus, it appears that the combination of aerobic training and strength training during MTP [22] adopted in our study may be a viable training strategy to reduce the serum levels of these lipid biomarkers. Longitudinal 8- [43,44] to 9- [32] months MTP interventions have been effective in improving the LP, showing reductions in TC and TG in older women. The reductions in the serum levels of TC and TG observed in our studies may be related to the increased activity of lipoprotein lipase in skeletal muscle [45], as well as the increase in capillary density and the potential removal and use of free fatty acids as a source of energy for the exercise [46,47], in addition, of course, to the increase in post-exercise energy expenditure associated with greater fat oxidation [48]. On the other hand, there was no change in %BF between baseline and the 8-week MTP. For this variable, Leite et al., [49] highlight that changes in decreasing the %BF and increasing the lean mass are more related to strength training than MTP. Although changes in the LP have been studied in longitudinal experimental designs [32,43,44], according to the present study, an eight-week MTP may be sufficient to promote decreases in serum levels of TC and TG in older woman.

Another continuous aspect associated with dependence and disability among older people is FC deterioration [50,51], leading to the inability to perform activities of daily living with satisfactory quality [52]. 30-CS has good properties for measuring lower-limb muscle strength in older people [53]. Our results are in line with studies that analyzed the MTP effect on 30-CS [54,55]. Additionally, we found an increase of approximately two repetitions after an eight-week MTP, which may be considered a significant clinical improvement for this population [56]. For the 8-FUG variable, although our results showed a statistical difference after an eight-week MTP regarding the baseline, the gross decrease in time in seconds was a less than clinically significant improvement (i.e., 3.4-s) [57]. The difference between the results may be partially explained by the reduced intervention time adopted in our study compared to other investigations that used MTP for longer periods (i.e., 12 weeks to six months) [58,59,60]. In the 6-MWT parameter, our results showed a significant increase in the distance covered during 6-MWT, which is consistent with other studies involving the MTP in older people [61,62]. Similar to our study, Suzuki et al., [62] found significant improvements in 6-MWT after a 56-week MTP intervention in OW. These results taken together show that significant improvements in the distance covered during a 6-MWT may be achieved when combining aerobic training and strength training [63]. In this line, Sousa et al., [64] showed that combined training (i.e., aerobic training + strength training) was more effective in improving the performance in the 6-MWT than aerobic training performed alone in older men. The HRPeak during the 6-MWT also increased significantly after the eight-week MTP. It is well established that the HRPeak declines with aging [65], and this decline may be related in part to the decreased sensitivity of the heart to beta-adrenergic stimulation, in addition to the decreased calcium flow associated with changes in the pacemaker tissue and the effect of prolonged diastolic filling [66]. Thus, there is consistent and congruent evidence in our study indicating that maintaining a high to moderate level of cardiorespiratory fitness may delay the decline in HRPeak with advancing age [65]. HGS is a usable measure of muscular strength [39] to quantify the static force that the hand can squeeze around a dynamometer [67], and is an important biomarker of health status in older people [68], due also to its association with FC [69]. Regarding the application of HGS after an MTP period in older people, the results are contradictory. In line with our study, Cadore et al., [70] also observed significant improvements in HGS after a 12-week MTP in nonagenarians. Courel-Ibáñez & Pallarés [71] also showed short-term benefits of MTP during HGS in frail and institutionalized older people, while Arrieta et al. [54] found no improvement in these parameters after a three-month MTP. Nonetheless, the differences among study results may be mainly attributed to MTP protocols, in which the program duration, as well as the combined physical capabilities (i.e., strength, endurance, balance and coordination), vary greatly [23].

Regarding detraining, most studies focused on verifying the detraining effects after MTP for uninterrupted periods corresponding to 1 year [21], 5-month [33], 3-month [21,29,72], 52- [73], 14- [74], 8- [36], and 6-week [73,74] on FC and LP in older people. However, only Toraman & Ayceman [34], from progressive cuts at two, four, and six weeks after a nine-week MTP period, assessed the main determinants of aerobic fitness, muscular strength, and some metabolic markers. Similarly, our study assessed FC and LP at weeks 2, 4, and 8. Assessing the younger older people, Toraman & Ayceman [34] found a significant decrease in the 8-FUG, 30-CS, and 6-MWT tests after week 4 and week 6 during detraining, compared to post-MTP. On the other hand, our study showed a significant worsening in the 8-FUG and 6-MWT tests regarding post-MTP from week 8 of detraining. However, 30-CS was the only parameter that remained unchanged in the present study throughout the 8-week of DT. Nevertheless, both studies did not demonstrate any significant change in the previously mentioned tests between week 2 and post-MTP. Ratel et al. [36] verified a significant decline in the maximal voluntary contraction strength of knee extensors, but did not observe any change in aerobic fitness after eight weeks of post-MTP detraining in older people. These results indicate that an eight-week post-MTP detraining period may partially reverse gains in aerobic fitness and completely abolish gains in muscular strength. As the tests assessed in our study and in the study by Toraman & Ayceman [34] are the result of the combination of aerobic fitness and muscular strength, it is possible that significant decreases in FC with detraining begin to be accentuated between week 4 and week 8. In general, studies involving post-MTP detraining over a fifty-two-week [73], three-week [29,72] and five-month [33] period also showed a significant decrease in performance on the 8-FUG [33,72,73], 30-CS [33,72,73], 6-MWT [29,33,72,73] tests. Additionally, in addition to the deleterious effects on FC, our study also observed significant declines in the LP (i.e., TC and TG) between week 8 of detraining and post-MTP, which translates into an increased cardiovascular risk [9]. Significant changes in BF% were also observed between the eight-week MTP and the four-week DT. Although Ratel et al. [36] did not observe changes in the LP after eight weeks of post-MTP detraining, other studies that assessed detraining for one year [21] and three months [21,29] showed results compatible with the present study. Furthermore, our study also observed a significant decrease in HRPeak and HGS in the eighth week of post-MTP detraining. The decrease in the HRPeak is a consequence of advancing age that may be minimized with physical exercise [65]. However, reductions in the HRPeak values as a result of age and physical inactivity may be concerning, due to their association with increased risk of stroke [75]. In the same way, decreases in HGS are also undesirable, since declines in performance on this test are related to cardiovascular, cancer, and all-cause mortality [76]. In summary, knowing that physical-training programs for the older people presents a typical detraining period of two to three months due to the summer vacation [33], the results of the studies together reveal the need for physical-education professionals to outline specific strategies for maintaining the adaptations acquired through the MTP. According to the results, these interventions to maintain acquired adaptations should begin between the fourth and eighth weeks after MTP interruption. Furthermore, other strategies, such as VIVIFRAIL may be recommended during this period of training interruption, due to its precise targeting of physical activity prescription for older people [77]. Another option is the possibility of including intermittent interventions with no more than ten weeks of inactivity between periods of exercise, which have already been shown to have a positive effect in protecting against functional decline in older people [74].

Furthermore, our study is not free from limitations. The main one is related to the non-inclusion of balance training in the MTP performed in the present study. Balance training is a very important component of MTP, as it prevents fall risk in older people [70]. Therefore, we encourage future research to include and assess balance training as one of the components of FC according to the Short Physical Performance Battery [78].

## 5. Conclusions

According to the present study, eight weeks of MTP is capable of significantly improving the LP (i.e., TC and TG), as well as the FC (i.e., HGS, HRPeak, 6-MWT, and 8-FUG) in OW. Furthermore, the adaptive responses in the LP and FC resulting from eight weeks of MTP may be maintained for a two- to four-week detraining period. From the eighth week of detraining, these parameters begin to decline significantly.

## Figures and Tables

**Table 1 healthcare-11-03021-t001:** Follow-up of detraining on the body fat percentage and lipid profile.

		EG		CG	
		Mean ± SD	95% CI	Mean ± SD	95% CI
%BF (%)	Baseline	39.20 ± 2.04	38.27–40.12	41.04 ± 2.01	39.88–42.21
	8-week MTP	38.74 ± 1.76	37.94–39.54	41.54 ± 1.88	40.38–42.71
	2-week DT	39.03 ± 1.92	38.16–39.91	43.02 ± 5.26	39.98–46.06
	4-week DT	39.26 ± 1.86 ^b^	38.42–40.11	39.60 ± 2.13	38.37–40.83
	8-week DT	39.28 ± 1.81	38.46–40.10	39.70 ± 2.25	38.40–41.00
TC (mg/dL)	Baseline	216.57 ± 5.22	214.20–218.95	219.79 ± 4.77	214.58–225.00
	8-week MTP	186.91 ± 5.07 ^a^	184.60–189.21	222.29 ± 11.57	216.80–227.78
	2-week DT	189.14 ± 5.69 ^d^	186.55–191.74	224.29 ± 4.65	219.98–228.59
	4-week DT	188.57 ± 6.22 ^e^	185.74–191.40	222.86 ± 6.20	219.70–226.01
	8-week DT	201.67 ± 5.68 ^c,f^	199.08–204.25	223.21 ± 5.87	220.39–226.04
TG (mg/dL)	Baseline	224.71 ± 4.95	222.46–226.97	231.64 ± 9.02	228.89–234.39
	8-week MTP	204.33 ± 5.18 ^a^	201.98–206.69	228.93 ± 9.51	222.25–235.61
	2-week DT	205.19 ± 5.6 ^d^	202.64–207.74	235.29 ± 7.46	232.60–237.97
	4-week DT	206.14 ± 6.91 ^e^	203.00–209.29	234.21 ± 5.46	229.63–236.80
	8-week DT	215.67 ± 6.51 ^c,f^	212.70–218.63	233.79 ± 4.89	230.40–237.18

EG (*n* = 21); CG (*n* = 14); %BF: Body fat percentage; TC: total cholesterol; TG: triglycerides; MTP: multicomponent training program; DT: detraining; ^a^ baseline vs. 8-week MTP, *p <* 0.05; ^b^ 8-week MTP vs. 4-week DT, *p <* 0.05; ^c^ 8-week MTP vs. 8-week DT, *p <* 0.05; ^d^ baseline vs. 2-week DT, *p <* 0.05; ^e^ baseline vs. 4-week DT; ^f^ baseline vs. 8-week DT.

**Table 2 healthcare-11-03021-t002:** Follow-up of detraining on functional capacity.

		EG		CG	
		Mean ± SD	95% CI	Mean ± SD	95% CI
HGS (kg)	Baseline	17.53 ± 1.58	16.81–18.25	16.73 ± 0.58	16.40–17.06
	8-week MTP	18.83 ± 1.74 ^a^	18.04–19.63	16.84 ± 0.62	16.49–17.19
	2-week DT	18.79 ± 1.75 ^c^	18.00–19.59	16.81 ± 0.71	16.44–17.18
	4-week DT	18.71 ± 1.77 ^d^	17.91–19.52	16.71 ± 0.79	16.34–17.09
	8-week DT	17.99 ± 1.69 ^c,f^	17.22–18.76	17.13 ± 1.39	16.32–17.93
HRPeak (bpm)	Baseline	131.48 ± 2.4	130.38–132.57	130.36 ± 2.84	128.72–132.00
	8-week MTP	135.48 ± 3.12 ^a^	134.05–136.90	130.64 ± 2.9	128.97–132.32
	2-week DT	135.00 ± 2.98 ^c^	133.64–136.36	130.36 ± 3.23	128.50–132.22
	4-week DT	134.86 ± 3.66 ^d^	133.19–136.53	129.36 ± 3.46	127.36–131.35
	8-week DT	132.24 ± 3.28 ^b^	130.74–133.73	131.43 ± 2.21	130.15–132.70
6-MWT (m)	Baseline	571.67 ± 48.8	549.46–593.88	553.57 ± 20.23	541.89–565.25
	8-week MTP	593.33 ± 41.2 ^a^	574.60–612.07	547.86 ± 20.46	536.04–559.67
	2-week DT	581.67 ± 39.9 ^c^	563.50–599.84	547.64 ± 21.04	535.50–559.79
	4-week DT	580.24 ± 40 ^d^	562.04–598.44	543.50 ± 25.88	528.56–558.45
	8-week DT	571.91 ± 37.8 ^b^	554.71–589.10	528.93 ± 34.09	509.25–548.61
8-FUG (s)	Baseline	5.82 ± 0.26	5.71–5.94	6.00 ± 0.35	5.80–6.20
	8-week MTP	5.52 ± 0.31 ^a^	5.39–5.66	6.07 ± 0.37	5.83–6.31
	2-week DT	5.68 ± 0.24 ^c^	5.57–5.80	6.04 ± 0.42	5.80–6.29
	4-week DT	5.67 ± 0.25 ^d^	5.55–5.78	5.92 ± 0.34	5.72–6.12
	8-week DT	5.83 ± 0.18 ^b^	5.74–5.91	5.93 ± 0.32	5.75–6.12
30-CS (rep)	Baseline	16.81 ± 2.14	15.84–17.78	17.07 ± 1.54	16.18–17.96
	8-week MTP	18.52 ± 3.22 ^a^	17.06–19.99	16.43 ± 1.81	15.35–17.51
	2-week DT	17.76 ± 2.76 ^c^	16.51–19.02	16.29 ± 1.68	15.31–17.26
	4-week DT	17.76 ± 2.41 ^d^	16.67–18.86	16.83 ± 2.03	15.88–17.46
	8-week DT	17.24 ± 2.21 ^e^	16.23–18.25	17.34 ± 1.87	16.35–18.51

EG (*n* = 21); CG (*n* = 14); HGS: handgrip strength; HRPeak: heart-rate peak in 6-min walk test; 6-MWT: 6-min walk test; 8-FUG: 8-foot up and go; 30-CS: 30 s chair stand; MTP: multicomponent training program; DT: detraining; ^a^ baseline vs. 8-week MTP, *p <* 0.05; *p <* 0.05; ^b^ 8-week MTP vs. 8-week DT, *p <* 0.05; ^c^ baseline vs. 2-week DT, *p <* 0.05; ^d^ baseline vs. 4-week DT; ^e^ baseline vs. 8-week DT.

## Data Availability

The data presented in this study are available on request from the corresponding author.

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
