# Peer review of "Follow-Up of Eight-Weeks Detraining Period after Exercise Program on Health Profiles of Older Women"

_healthcare, 2023, doi:10.3390/healthcare11233021_

Round 1

Reviewer 1 Report

Comments and Suggestions for Authors

“Follow-up of 8-weeks Detraining Period after exercise program on health profiles of older women” by Leitão and colleagues aimed at investigating the 8-week multicomponent training program (MTP) effects followed by 2-, 4- and 8-week of interruption on the lipid profile (LP) and functional capacity (FC) of older women.

Although the topic is of current interest to the scientific community, the manuscript needs minor revisions before publication in Healthcare.

Abstract – This section is of paramount importance because it should entice the reader to read the work in full. In this case, although the authors have described the content of their manuscript in detail, it is in my opinion incomplete. In fact, it would be appropriate to add a concluding sentence suggesting possible future perspectives that might emerge from the results obtained.

Introduction -  This section should be improved. In the middle section, the authors refer to the effects of a multicomponent training (MTB) program on cardiovascular health and physical functioning in older adults. I believe that this part needs to be deepened with quantitative as well as descriptive data so that the reader can understand the assumptions of the work done. 

Comments on the Quality of English Language

Minor editing of English language required

Author Response

Firstly, we would like to thank you for your comments, which will improve the quality of our manuscript. We will respond item by item and leave the lines and text referring to the requested changes as a response. Note: Changes to the main text are in red.

“Follow-up of 8-weeks Detraining Period after exercise program on health profiles of older women” by Leitão and colleagues aimed at investigating the 8-week multicomponent training program (MTP) effects followed by 2-, 4- and 8-week of interruption on the lipid profile (LP) and functional capacity (FC) of older women.

Although the topic is of current interest to the scientific community, the manuscript needs minor revisions before publication in Healthcare.

Abstract – This section is of paramount importance because it should entice the reader to read the work in full. In this case, although the authors have described the content of their manuscript in detail, it is in my opinion incomplete. In fact, it would be appropriate to add a concluding sentence suggesting possible future perspectives that might emerge from the results obtained.

A: We add.

Introduction -  This section should be improved. In the middle section, the authors refer to the effects of a multicomponent training (MTB) program on cardiovascular health and physical functioning in older adults. I believe that this part needs to be deepened with quantitative as well as descriptive data so that the reader can understand the assumptions of the work done.

A: We add more sentences in introduction.

Reviewer 2 Report

Comments and Suggestions for Authors

Authors present an interesting manuscript on multicomponent training and exercise cessation in older women. Results of this research would be of great practical value for the daily practice of strength and conditioning trainers in geriatrics. Congratulations for increasing the knowledge on this important topic. However, several aspects throughout the manuscript should be addressed, previously to acceptance.

 GENERAL COMMENTS

Please ask a native speaker to double-check the writing.

 SPECIFIC COMMENTS

INTRODUCTION:

- Line 51: Multicomponent training program is abbreviated as MTB, when previous line is reported as MTP. I supposed that it is a mistake, please review it.

- Line 52: The balance training should be included in MTP as an essential component of this methodology and very important for preventing falls (DOI: 10.1007/s11357-013-9586-z ). Static balance capacity can be included even in functional capacity according to the Short Physical Performance Baterry (DOI: 10.1093/geronj/49.2.m85 ).  Although static balance have not been evaluated in this papers, I encourage the authors to perform it in future studies.

- Line 56 and 65: In this sentence’s authors explain some reason for interrupting a exercise routine (physical factor and vacations). Although both factors are real in training context, it should be noted that exercise interventions in older adults are commonly interrupted due to falls, illness, or hospitalizations (https://doi.org/10.1093/gerona/glr142).

-Line 57: “Residual effect” may be more appropriate and generalized than the term “level of retention”. (https://doi.org/10.1111/sms.14428)

- Line 56: Vacation period should not be a real reason of exercise cessation. There are many options to conduct unsupervised routines in older adults (example: VIVIFRAIL training program from Mikel Izquierdo).

 METHOD:

- Line 75: “older women with functionally independent” according to what criteria? Are these women living in the community (i.e., community-dwelling older adults)? If so, please consider it.

- Line 77: “The exclusion criteria were: a) have already participated in any regular physical  activity program training”  Never???

- Line 90: “The participants were separated into two groups” This randomization must be done based on some criteria. What criteria were used?

- Line 90-92: Why the number of participants in both groups are so different? Did you conduct a sample size calculation?

- Line 101: MTP program and monitoring (exercises, volume, intensity, rest…) need more details.

-Line 136-137: Sit-to-stand, 6-min walking and time up & go tests need more details.

- Line 143. Handles positioned should be standardized according to sex in older adults. If this was not performed, it was a methodological flaw that should be taken into consideration, even though I imagine that the same grip was maintained pre-post intervention. These data would be of dubious use if they were to be used for normal values for this population.

- Line 145:  If you record heart rate, why do not use mean heart rate in the analyses?

-Line 155: Effect size should be calculated using Hedge’s g when the research included small sample size per group and gerontology (doi: 10.1093/geroni/igz036).

- Line 156: Interpretation small, moderate, or large of ES, should be referenced.

RESULTS:

- Could you inform the dropout of intervention and adherence to the exercise program?

- A table should be included in results section analyzing the independence of groups at baseline. This also is needed in statistical analysis with an “t-test”.

- This research presents a well analysis of training and detraining effects on body fat, lipid profile and functional capacity. However, the information in the table seems cluttered and difficult to understand. This tables should be restructured to make them easier for the reader.

DISCUSSION:

- Line 188: “the CG did not change their parameters after 8-week”. The values changed, although no benefits were shown.

- Line 199: “Robust reviews” is an unclear concept.

-Line 219: What are psychometrics properties? P.S: review the writing. Did you mean capability to measure muscle strength?

-Line 224 and 230-231: Please use these sentence in methods.

-Line 251: Other studies show short-term benefits on HGS in frailty and institutionalized older adults (DOI: https://doi.org/10.1016/j.jamda.2020.11.007).

-Line 257: Functional capacity preservation after MTP cessation in short and long-term have been analyzed in older adults (https://doi.org/10.1016/j.jamda.2021.05.037). This data can be discussed supporting our results. Moreover, I encourage the authors to use “protective effect of exercise” concept when the value after exercise cessation period is significantly better than baseline.

- Line 269: Please change “however” because it was used in previous sentences. Maybe nevertheless can be used.

- Line 292-296: In this sentences can be included the possibility of “Intermittent interventions with no more than X weeks of inactivity between exercise periods can be appropriate and sufficient to protect older adults from functional decline” This idea is supported by other studies (https://doi.org/10.1016/j.jamda.2021.05.037).

Comments on the Quality of English Language

Please ask a native speaker to double-check the writing.

Author Response

Firstly, we would like to thank you for your comments, which will improve the quality of our manuscript. We will respond item by item and leave the lines and text referring to the requested changes as a response. Note: Changes to the main text are in red.

Introduction

Line 51: Multicomponent training program is abbreviated as MTB, when previous line is reported as MTP. I supposed that it is a mistake, please review it.

A.: This was our mistake. We changed the abbreviation to MTP throughout the manuscript.

Line 52: The balance training should be included in MTP as an essential component of this methodology and very important for preventing falls (DOI: 10.1007/s11357-013-9586-z ). Static balance capacity can be included even in functional capacity according to the Short Physical Performance Battery (DOI: 10.1093/geronj/49.2.m85 ).  Although static balance has not been evaluated in this papers, I encourage the authors to perform it in future studies.

A.: We will include balance training in our future studies. Furthermore, we consider the lack of balance training as a limitation of our study.

Line 56 and 65: In this sentence’s authors explain some reason for interrupting a exercise routine (physical factor and vacations). Although both factors are real in training context, it should be noted that exercise interventions in older adults are commonly interrupted due to falls, illness, or hospitalizations (https://doi.org/10.1093/gerona/glr142).

A.: We added the suggested reference and changed the sentence

Line 57: “Residual effect” may be more appropriate and generalized than the term “level of retention”. (https://doi.org/10.1111/sms.14428)

A.: We changed the term and added the suggested reference

Line 56: Vacation period should not be a real reason of exercise cessation. There are many options to conduct unsupervised routines in older adults (example: VIVIFRAIL training program from Mikel Izquierdo).

A.: We agree it shouldn't be, but it is. That's why we leave a recommendation about this supervised training option at the end of the discussion.

Method

Line 75: “older women with functionally independent” according to what criteria? Are these women living in the community (i.e., community-dwelling older adults)? If so, please consider it.

A: We added community-dwelling older women. (LINE 85)

- Line 77: “The exclusion criteria were: a) have already participated in any regular physical  activity program training”  Never???

A: We add in the last 12 months.(LINE 89)

- Line 90: “The participants were separated into two groups” This randomization must be done based on some criteria. What criteria were used?

A: We used a simple random sampling. We add sentence.(LINE 101)

- Line 90-92: Why the number of participants in both groups are so different? Did you conduct a sample size calculation?

A: We add “In the CG six older women were excluded because they did not attend all assessments..”and “Sample size calculation was made with G*Power program considering a 0.3 effect size, 83% of power and p<0.05 of significance, a sample of 21 older women per group would be necessary.” (LINE 104-105; LINE 178-180)

- Line 101: MTP program and monitoring (exercises, volume, intensity, rest…) need more details.

A: We add details and references to support. (LINE123-132)

-Line 136-137: Sit-to-stand, 6-min walking and time up & go tests need more details.

A: we already have the reference to support this. Since these are the most common battery tests used in older adults we do not add any test description. Although, we added like you suggested.  (LINE152-166)

- Line 143. Handles positioned should be standardized according to sex in older adults. If this was not performed, it was a methodological flaw that should be taken into consideration, even though I imagine that the same grip was maintained pre-post intervention. These data would be of dubious use if they were to be used for normal values for this population.

A: We rewrite and add sentence to support. (LINE159-164)

- Line 145:  If you record heart rate, why do not use mean heart rate in the analyses?

A: We accept the recommendation but we only wanted to consider the peak.

-Line 155: Effect size should be calculated using Hedge’s g when the research included small sample size per group and gerontology (doi: 10.1093/geroni/igz036).

A: We accept the suggestion but we considered cohen effect size like many other studies.

- Line 156: Interpretation small, moderate, or large of ES, should be referenced.

A: We have the interpretation in the statistical analysis sub-section.

Results

- Could you inform the dropout of intervention and adherence to the exercise program?

A: We add sentence about dropout in the methods section and adherence to the program in results section(“The Adherence of EG was 91%.”). (LINE185)

- This research presents a well analysis of training and detraining effects on body fat, lipid profile and functional capacity. However, the information in the table seems cluttered and difficult to understand. This tables should be restructured to make them easier for the reader.

A: We changed. (LINE 191; 207)

Discussion

Line 188: “the CG did not change their parameters after 8-week”. The values changed, although no benefits were shown.

A.: The values were not changed significantly. Therefore, we do not include it in the discussion.

Line 199: “Robust reviews” is an unclear concept.

A.: We rewrite to reviews.

Line 219: What are psychometrics properties? P.S: review the writing. Did you mean capability to measure muscle strength?

A.: We have withdrawn this term.

Line 224 and 230-231: Please use these sentence in methods.

A.: We rewrite.

Line 251: Other studies show short-term benefits on HGS in frailty and institutionalized older adults (DOI: https://doi.org/10.1016/j.jamda.2020.11.007).

A.: We added the suggestion. “…Courel-Ibáñez & Pallarés [71] also showed short-term benefits of MTP during HGS in frail and institutionalized older people…” (LINE278-279)

Line 257: Functional capacity preservation after MTP cessation in short and long-term have been analyzed in older adults (https://doi.org/10.1016/j.jamda.2021.05.037). This data can be discussed supporting our results. Moreover, I encourage the authors to use “protective effect of exercise” concept when the value after exercise cessation period is significantly better than baseline.

A.: We add the requested reference.

Line 269: Please change “however” because it was used in previous sentences. Maybe nevertheless can be used.

A.: We changed. (LINE295)

Line 292-296: In this sentences can be included the possibility of “Intermittent interventions with no more than X weeks of inactivity between exercise periods can be appropriate and sufficient to protect older adults from functional decline” This idea is supported by other studies (https://doi.org/10.1016/j.jamda.2021.05.037).

A.: We changed. “… Another option is the possibility of including intermittent interventions with no more than 10-week of inactivity between periods of exercise, which have already been shown to have a positive effect in protecting against functional decline in older people [74]…” (LINE324-329)

Reviewer 3 Report

Comments and Suggestions for Authors

Dear Authors,

Congratulations for the work done, I have some suggestions/questions that can improve the article.

Abstract.

To add the meaning of all abbreviations, EG (experimental group), CG,…. BMI (introduction)….

Conclusion must be corrected; the study did not know what happen at 5,6 or 7 week, the collected data are from 2, 4 and 8 weeks of detraining

Keywords.

Do not repeat words in the title, and it will be better to present by alphabetic order.

Introduction

To correct “multicomponent training program (MTB)” in all document

MCT also include balance, so it can be included in the text.

lipid profile, functional capacity, older women… have abbreviation in the abstract, as this words are repeated several times in the document to add the abbreviation can be practical

Material and Methods

There is no information about the training program (Volume, intensity, set, rep, recovery…..) it must be added

Do not repeat meaning of abbreviations, TC, TG….

To be uniform p<0.05 or p<.05

Table 1 and 2. The significant effect (a,b,c…) could be presented in bold for readers to see better. Beside, to reduce at one decimal should also be included in those variables where is possible.

Limitations should be included in the article. Participants were randomly in the two groups? Training program was controlled? ….

Best Regards

Author Response

Firstly, we would like to thank you for your comments, which will improve the quality of our manuscript. We will respond item by item and leave the lines and text referring to the requested changes as a response. Note: Changes to the main text are in red.

Abstract

To add the meaning of all abbreviations, EG (experimental group), CG,…. BMI (introduction)….

A: We made the recommended changes.

 Conclusion must be corrected; the study did not know what happen at 5,6 or 7 week, the collected data are from 2, 4 and 8 weeks of detraining.

A: We changed

Do not repeat words in the title, and it will be better to present by alphabetic order.

A: We made the recommended changes.

Introduction

To correct “multicomponent training program (MTB)” in all document.

A: We corrected this mistake throughout the manuscript.

MCT also include balance, so it can be included in the text.

A.: We prefer to put the non-use of balance as a limitation of our study (please check the end of the discussion for more details).

lipid profile, functional capacity, older women… have abbreviation in the abstract, as this words are repeated several times in the document to add the abbreviation can be practical.

A.: We have added abbreviations throughout the manuscript.

Material and Methods

There is no information about the training program (Volume, intensity, set, rep, recovery…..) it must be added

A: We changed.

Do not repeat meaning of abbreviations, TC, TG….

A: We changed.

To be uniform p<0.05 or p<.05

A: We uniformized.

Table 1 and 2. The significant effect (a,b,c…) could be presented in bold for readers to see better. Beside, to reduce at one decimal should also be included in those variables where is possible.

A: We changed.

Limitations should be included in the article. Participants were randomly in the two groups? Training program was controlled? ….

A: We added.

Round 2

Reviewer 2 Report

Comments and Suggestions for Authors

The revision work is good, and I think this paper can be accepted for publication.

Comments on the Quality of English Language

The revision work is good, and I think this paper can be accepted for publication.